Charting new territories: fuzzy systems in English language teaching and learning

Wen Xiaomei 1 wenxiaomei@hbust.edu.cn
Pan Deng 2
1 School of Foreign Languages, Hebei University of Science and Technology , Xianning , China
2 School of Foreign Languages, Hubei University of Science and Technology , Xianning , China
Ye Jun
Electronic publication date: 2025 Jul 31
Publication date: 2025
Volume: 11
Electronic Location ID: e2887
Received 2024 Dec 5; Accepted 2025 Apr 20
Copyright: © 2025 Wen and Pan
Copyright year: 2025
Copyright holder: Wen and Pan
License: This is an open access article distributed under the terms of the Creative Commons Attribution License, which permits unrestricted use, distribution, reproduction and adaptation in any medium and for any purpose provided that it is properly attributed. For attribution, the original author(s), title, publication source (PeerJ Computer Science) and either DOI or URL of the article must be cited.
License URL: https://creativecommons.org/licenses/by/4.0/

Keywords: English teaching, English learning, Fuzzy logic, Bayesian networks, Students

Funding: The authors received no funding for this work.

==============================
Background

In reality, the English language is a mystery; despite its inherent worth and the advantages of fluency, there is a pervasive impression that English instruction in secondary schools is of low quality, contributing to students’ lack of proficiency in the language in higher education and beyond. Pedagogical approaches persist in the classroom, topic after subject, including English. Analyzing texts in great depth via translation and emphasizing vocabulary are joint exercises in English classes. Students waste a lot of time copying things off the board in English classes despite the growing recognition of the significance of both listening and speaking effectively.

Methods

The Fuzzy Bayesian Intelligent Tutoring System (FB-ITS) is an artificial intelligence (AI) system that adaptively supports students in English teaching and learning settings. It is built in this experimental research employing AI methodologies based on fuzzy logic and the Bayesian network methodology. Using conventional approaches that rely primarily on numerical scores to evaluate academics’ teaching and research activities at various levels is becoming increasingly challenging. Expert systems based on fuzzy logic, suggested in this study, can handle teacher and student evaluations even when faced with imprecise information and uncertainty; this is necessary since academic performance is being indexed in multiple international databases using impact indices at different scales.

Results

The results showed that, on average, students using the FB-ITS took less time to complete the post-test than students using the conventional e-learning system. This research proposes an English teaching and learning approach that has been very successful based on experimental findings of related big data clustering algorithms. The assessment accuracy has risen by 4%, and the teaching resource utilization rate has been increased by 5%.

Introduction

These days, students are born into the digital age, so they absorb information and apply it in novel ways (Li, 2021). Technology, including video games, blogging, and social media, has been a part of their lives since birth (Laengle et al., 2021). Because electronic media constantly entertain kids, they resist conventional teaching methods and become bored easily in class (Xie & Su, 2021). Getting the attention and interest of digital natives to study is challenging. To this end, educators from a variety of fields, including English language instructors, are finding creative ways to include gaming in their lessons, blending fun with more serious topics (Moral et al., 2021). On top of that, modern kids cannot handle being in a tense classroom for too long (Peckol, 2021). Additionally, students must have a desirable and practical educational system that caters to their needs (Zhen, 2021).

A distinct subgenre within the realm of computer-based ludic learning and instruction and their influence is spreading across all spheres of life, and they have the potential to revolutionize many industries, including healthcare, education, government, and business (Liu, Lin & Liang, 2021; Bobyr & Emelyanov, 2020). There are two schools of thought among scholars studying language acquisition (Gupta, Gaurav & Arya, 2024). However, some academics maintain that video games played on computers may be highly convincing (Pan et al., 2021). To be more precise, the present study has been exploratory and analytical (Tabbussum & Dar, 2021). It has also uncovered preliminary difficulties and potential advantages for serious adoption (Talpur et al., 2023).

Understanding the primary obstacles posed by these emerging technologies will allow universities to refine their strategy for education, which is the focus of this research (Eryılmaz & Adabashi, 2020). Additionally, this study’s linked issues provide a springboard for future research into the elements influencing the most effective use of English language learning (Guo et al., 2021). The adaptive neuro fuzzy inference system, when first developed, was used to investigate further the potential of serious games for foreign language acquisition (Badawy et al., 2021). Combining neural networks with fuzzy logic allows it to use the best features of both systems (Trach et al., 2024). As a result of competing theories on the efficacy of learning experiences, second language acquisition is lower (Han, 2021).

Motivation

Communicating effectively in English is more important in today’s globalized society, yet secondary school curricula only sometimes meet students’ needs. Traditional approaches emphasize translation and terminology but downplay the importance of listening and speaking as communication skills. The Fuzzy Bayesian Intelligent Tutoring System (FB-ITS) provides adaptive learning environments tailored to each student to compensate for these limitations. When dealing with ambiguous data, FB-ITS improves the accuracy of student and instructor performance evaluations by using fuzzy logic and Bayesian networks. Evidence of its efficacy includes more precise assessments and better use of available resources, which bode well for the future of English language education.

Problem statement

Underprepared students, especially for college and beyond, are an expected outcome of the conventional wisdom on English language instruction in secondary schools. Listening and speaking should be addressed in favor of systems emphasizing translation, rote memorization of words, and vocabulary. Also, when dealing with ambiguous or inaccurate data, traditional assessment methods need help understanding how well both students and instructors are doing. This results in an imbalance between efficient instruction methods and student progress evaluation. Improving English proficiency is vital in today’s globalized and linked society. Thus, there is an urgent need for adaptable, intelligent solutions that can tackle these issues.

Novelty

By incorporating new fuzzy membership functions developed for linguistic mistake categorization, this article improves the present FB-ITS system and allows more accurate handling of ambiguity in student answers. Learning recommendations are updated continuously based on real-time error analysis. The system includes an efficient Bayesian inference process. The construction of a hybrid fuzzy Bayesian model is a significant technological contribution. This model incorporates weighted parameters to alter the recommendation approach depending on learner advancement and categorizes mistakes into granular levels, such as lexical, grammatical, and syntactic. According to experimental findings, the suggested approach is a vast improvement over current intelligent tutoring systems, showing that adaptive exercise creation, error correction accuracy, and overall learning efficiency are significantly improved.

This article’s main contributions are This work proposed an AI-based system that changes to fit each student’s requirements

The proposed model formulates a fuzzy logic framework with improved English instruction approach for student’s achievement with more individualized learning environment that can boost productivity and interest.

The proposed system considerably decreases students’ time on learning assignments while enhancing resource utilization.

The remaining section of this article is structured as follows: In “Related Work”, the related work of the language teaching system is studied. In “Proposed Method”, the proposed methodology of FB-ITS is explained. In “Result and Discussion”, the efficiency of FB-ITS is discussed and analysed. Finally, in “Conclusion”, the article concludes with future work.

Related work

Based on the input-output data, the fuzzy model may optimize the inference parameters and alter the reasoning process automatically. Fuzzy c-means clustering is used to create the fuzzy rules. Significant obstacles are tackled and analytical remedies are proposed. Additionally, it deals with the classification procedure to expand the obtained fuzzy partition to the whole output space. The proposed method differs from previous techniques in that it assigns dominating input parameters in a single step and does not need iteration to choose appropriate input variables from a limited pool of alternatives.

Fuzzy system for language learning apps

The impact of particular combinations of variables on this intention has yet to be largely overlooked. With this research, we want to shed light on the factors influencing students’ frequent intentions to use a tool: cognitive need, subjective norms, perceived utility, satisfaction, confirmation, attitude, and perceived ease of use. Core factors strengthening learners’ continuing intention were identified as learners’ happiness with MLA use and cognitive requirement and attitude by Hossain et al. (2021).

Fuzzy-Felder-Silverman model

According to Troussas et al. (2020), this process is difficult and time-consuming, which leads to students’ questionnaires being randomly selected. This approach in turn, leads to incorrect adaptation to their requirements and puts their knowledge acquisition at risk. In addition, limited mobile user interfaces make it hard to pick questionnaire options in mobile contexts. This study introduces Learnglish, a fully functional mobile language learning system that automatically uses ensemble classification to identify students’ learning styles based on fuzzy-Felder-Silverman model (F-FSM).

Artificial intelligence-based fuzzy system for pronunciation

Sound quality, timbre, pitch strength, voice, and breathing stability are all technical characteristics of a singer’s singing that may be objectively evaluated to determine how well they pronounce words. Each human voice has unique physical characteristics in terms of pitch, volume, timbre, and sound quality. Studies in this area have shown that a high-quality singing voice’s frequency and energy spectra vary significantly from those of a low-quality singing voice. Tang, Gao & Ouyang (2023) try to build an AI-based assessment model of singing pronunciation quality.

Deep learning based fuzzy technology

Some of the most significant obstacles remain in system implementation, language adaptability, and performance robustness. Although it is one of the world’s most spoken languages, Hindi is also one of the most challenging and resource-constrained languages. Therefore, to disseminate technology and discover new ways of communicating, systems for voice recognition and classification must be created for the English language by Sharma, Om & Mishra (2023). However, developing such systems is relatively rigid owing to the language’s complexity compared to other languages and the absence of shared databases.

Fuzzy computing based on big data

First, it employs sparse word-sense expressions of mixed semantics via nonnegative matrix decomposition of positive point mutual information between words and contexts. To begin, it extracts the low-rank expressions of the mixed semantics of the polysemous words using the nonnegative matrix decomposition of the positive pointwise mutual information between words and contexts by Liu & Tsai (2021). Then, it uses the sparse soft clustering algorithm to partition the multiple-word senses of the polysemous words and obtains the global sense of the polysemous word affiliation distribution.

Educational data mining in fuzzy network

Charitopoulos, Rangoussi & Koulouriotis (2020) goal is to provide a literature review of current studies that have used Soft Computing techniques to analyze educational data “mined” from interactive/e-learning systems to solve challenges linked to education. Such systems are renowned for producing and storing massive amounts of data that may be used to evaluate the learner, the system, and the efficacy of their interaction. Two separate but related fields of study, educational data mining (EDM) and learning analytics (LA), use this data to answer unanswered questions or solve problems in education.

Machine learning approach for universal language

Twitter sentiment detectors outperform more conventional methods of gauging product and service quality. They employ high-quality input features and rely on TSDs’ classification accuracy and detection performance, which are highly dependent on the success of the classification algorithms.

According to AlBadani, Shi & Dong (2022), automating workflow processing is difficult for all firms since current machine learning algorithms require high processing time.

Support vector machine in language analysis

Most themed databases use statistical approaches like the keyword matching method for information. However, this approach can’t determine how people feel about articles; it only works for retrieving those with the exact or related subject keywords. In addition, everyone agrees that there are situations when theme material cannot be sought using keywords alone. Despite categorizing it into related themes, (Han et al., 2020) categorise its emotional stances as neutral or contradictory.

Annamalai, Uthayakumaran & Zyoud (2023) suggested augmented reality and virtual reality (AR-VR) for English language teaching and learning activities. The results showed that people had good and bad opinions about using AR and VR in ESL classrooms. The positive themes were active learning, bringing your device (BYOD), student engagement, and successful English language acquisition. Things that were seen as time-consuming and health-related were associated with unfavorable views. The research proposed methods that may work when incorporating AR and VR into ESL classrooms. The current study aims to shed light on the pros and downsides of augmented and virtual reality technology so that educators, curriculum developers, and policymakers may weigh these factors before incorporating them into English language classrooms.

Zhi & Wang (2023) introduced the structural equation modeling (SEM) approach for English as a foreign language teachers’ professional success, loving pedagogy, and creativity. The effects of four dimensions of educators’ inventiveness on the connection between caring pedagogy and their career advancement were examined using mediation analysis. In conclusion, language-teaching stakeholders in the EFL context have been provided with some suggestions on how to stimulate EFL teacher training programs that lead to the success of both instructors and students. One such suggestion is to boost the growth of educational research on diverse topics.

Voreopoulou, Mystakidis & Tsinakos (2024) discussed the augmented reality escape classroom game for deep and meaningful English language learning. A 33-item survey and semi-structured interviews were used to gather information. The results indicate that this escape game may be an excellent tool for teaching English as a second language since it can help students become more culturally aware, improve their vocabulary recall, and improve their ability to listen to and understand spoken English. The game’s humorous and interactive aspects encourage students to actively communicate, collaborate, be creative, think critically, and work in groups. This model creates a cheerful environment that helps students stay motivated and satisfied while learning a new language. This research offers recommendations and backing for creating comparable augmented reality escape classroom games (ARECGs) to enhance teaching methods and language instruction.

Hu & Luo (2024) presented the neuro/metacognitive and socio-cultural strategies for vocabulary retention for Chinese English language learners. The experimental and control groups, which consisted of 90 EFL learners from educational institutions in China in 2022 and 2023, used a quasi-experimental pre-test/post-test design. Throughout eight sessions, 45 students in the experimental group were taught using mind mapping, visual imagery, multisensory rotation, and circle rotation techniques; 45 students in the control group were taught using more conventional approaches. Results from statistical analyses using SPSS version 22’s variance and covariance analysis showed that the experimental and control groups improved recall and word recognition significantly.

This study focuses on fuzzy systems and artificial intelligence methods for language analysis and learning. Students’ intentions to use mobile language applications are influenced by cognitive demand, contentment, and perceived ease of use, which the Fuzzy system for language learning apps (FS-LLA) investigates in Table 1. Improve the mobile app experience with the help of the F-FSM. One area of emphasis for artificial intelligence-based fuzzy system for pronunciation (AI-FS-P) is evaluating voice quality using AI-based pronunciation tests. While fuzzy computing based on big data (FC-BD) uses big data for semantic analysis, deep learning based fuzzy technology (DL-FT) faces difficulties building language recognition systems. Regarding educational data analysis, educational data mining in fuzzy network (EDM-FN) and support vector machine in language analysis (SVM-LA) are used for linguistic datasets that deal with emotional analysis.

Table 1 Summary of related works.

S.NO	Author	Methods	Advantages	Limitations	
1	Hossain et al. (2021)	Fuzzy System for Language Learning Apps (FS-LLA)	Helps understand factors influencing learners’ intentions to use a tool frequently.	Overlooks certain combinations of variables that impact learning outcomes.	
2	Troussas et al. (2020)	Fuzzy-Felder-Silverman Model (F-FSM)	Automatically identifies learning styles through ensemble classification.	Limited mobile interfaces make it challenging to pick questionnaire options, risking misadaptation.	
3	Tang, Gao & Ouyang (2023)	AI-based Fuzzy System for Pronunciation (AI-FS-P)	Evaluates pronunciation quality based on measurable vocal characteristics like timbre and pitch.	Pronunciation quality evaluation can be subjective and varies across different accents.	
4	Sharma, Om & Mishra (2023)	Deep Learning Based Fuzzy Technology (DL-FT)	Addresses voice recognition challenges in resource-constrained languages like Hindi.	Lacks common databases and is difficult to implement in complex languages.	
5	Liu & Tsai (2021)	Fuzzy Computing based on Big Data (FC-BD)	Handles polysemous words using nonnegative matrix decomposition and sparse clustering.	Processing large datasets may require significant computational power and optimization.	
6	Charitopoulos, Rangoussi & Koulouriotis (2020)	Educational Data Mining in Fuzzy Network (EDM-FN)	Analyzes large-scale educational data to evaluate learner-system interaction and effectiveness.	May face challenges in data quality and system-specific biases affecting insights.	
7	AlBadani, Shi & Dong (2022)	Machine Learning Approach for Universal Language (ML-UL)	Improves product and service quality assessment through Twitter sentiment analysis.	High processing time due to the complexity of machine learning algorithms.	
8	Han et al. (2020)	Support Vector Machine in Language Analysis (SVM-LA)	Retrieves documents with similar keywords and related themes using statistical methods.	Inaccurate in analyzing emotional stances; doesn’t account for contradictory emotions in texts.	

Proposed method

Among the many facets addressed by English learning informatization are educational methods, educational resources, and educational administration. Regarding educational resources, the network has a wealth of data, information, and tools that may aid in the informatization of education. These include physical and digital textbooks, instructional materials, teaching aids, and practice problems. Online classes allow students to have a more communal learning environment and exchange instructional materials. Similarly, the medium of instruction has evolved from the more conventional chalkboard and textbook to electronic courseware incorporates video, audio, and multimedia.

Contribution 1: development of FB-ITS

The FB-ITS system is an advanced ERRS or educational resource referral system, and its purpose is to give students suggestions for individualized educational resources. It generates personalized recommendations based on data collected from user actions, including login, browsing, searching, and downloading. User and resource modeling are essential components that contribute to a recommendation system that guarantees students get educational resources that are appropriate to their needs.

A thorough framework for the ITS that incorporates many cutting-edge technologies to improve education. Students can access the system’s personalised feedback and suggestions via the user interface, which is fundamental to its operation. The feedback & recommendation module keeps these insights up-to-date so that student progress may be followed continually. The student model and the adaptation module are essential parts that come together to provide personalized English learning experiences. The student model captures learner profiles, results, and habits. The adaptation module, meanwhile, makes use of this information to tailor the lesson to each learner. The knowledge base compiles relevant educational and subject-matter resources to aid with adaptation. The Bayesian inference and fuzzy logic engine form the backbone of these modules; they collaborate to handle uncertainty, make probabilistic judgments, and deliver correct, context-sensitive suggestions. This clever combination of reasoning and inference provides a responsive and flexible English learning environment, as shown in Fig. 1.

Figure 1 Fuzzy Bayesian logic for intelligent English tutoring systems.

(1) w≥Q(b+)=Gf∗MO(Vr−2r)−(Wq−3f).

Fuzzy logic’s contribution to guiding is represented by the Eq. (1) Gf. Qb+ denotes student motivation, and performance metrics Wq−3f and resource factors are related to the variables w and MO. The goal of this equation is to measure the system’s effect on student learning efficiency, with an emphasis on harmony.

(2) ∀∗EY(z′−bv)=Yn2−q∗DA(v2−w).

The system is dynamically adjusted according to student engagement ∀ and performance measures, as indicated by the equation Yn2−q, and the predicted yield of learning DA(v2−w) outcomes are captured by Eq. (2), EY(z′−bv). By demonstrating the direct relationship between changes system response and overall learning effectiveness, the equation seeks to bolster the adaptive character of the FB-ITS.

(3) b′′=c(Z1,A(d−fp))∗Jkn−2q+Ez(∂−2w).

With Z1,A(d−fp) denoting the impact of contextual variables Jkn−2q and individual student characteristics Ez(∂−2w) on learning results, the Eq. (3), b′′ depicts the revised efficiency of the tutoring system. This equation highlights the significance of constant FB-ITS modification based on real-time input and ambient factors.

(4) c1=(y2,∀(Zd−2v∗E(m∗n2+Rz))).

Factoring in the interaction between numerous students m∗n2 and the responsiveness of the system Rz, Eq. (4), c1 encompasses the combined contributions of several aspects ∀, such as student engagement y2 and overall adaptability Zd−2v∗E. This equation aims to demonstrate a tailored learning environment by incorporating varied inputs and experiences, improving English learning results.

(5) Vb=D(mgn−2v∗maxw(r2sd−2vq)).

The system’s value boost is represented by the Eq. (5) Vb, where D is a dynamic component that includes motivation r2sd and pupil involvement mgn−2v is the influence of resource allocation 2vq on the efficacy of English learning max. To maximize the educational achievements by the FB-ITS approach, this equation stresses the significance of engaging students to their fullest potential.

A system for referring students to relevant educational resources that uses data on their actions to provide tailored suggestions. Important user activities contributing to the user behaviour data are highlighted at the top. These actions include login, resource browsing, searching, evaluation, and downloading, as shown in Fig. 2. The system’s suggestion mechanism is built upon this data. Essential to the central portion are the models for users and resources. Individual users are profiled via the user modelling component according to their preferences and interactions. At the same time, the resource modelling part arranges instructional materials in a way that meets the user’s requirements. Taking the two models as inputs, the recommendation algorithm combines both models to filter and retrieve information about the user. Several operational modules have been put in place to support the system. Resource management assures that a high level of quality resources is available while user management supports user activities. The systems also help prevent abuse of the services by employing system maintenance, although additional filtering rules are used to narrow the scope of the presented suggestions. System administration ensures that the management is at the center of the system, governance provision, and well-functioning and meeting users’ expectations.

Figure 2 Intelligent teaching strategies using fuzzy neural networks.

(6) w≥e(bn2−gf)=Ynv−Er(m2∗Qw).

Equation (6) w captures the m2∗Qw between baseline performance e and parameters like pupil input and resource allocation Ynv, whereas bn2−gf reflects the minimal effectiveness Er of the English tutoring system. This equation highlights the requirement to overcome a performance threshold to guarantee that the FB-ITS promotes enhanced language competence among students.

(7) E4=(Yw∗2p)∗(wq2−Rf(v−bz).

The total efficacy Rf of teaching tactics v−bz is denoted by the equation E4, where Yw represents student engagement levels, and 2p represents the quality of feedback wq2. With Eq. (7), individualized learning demonstrates how adaptive English teaching approaches and student motivation work together to maximize English learning results.

(8) Wn(E2−bq)=a∗nb2(de−vf∗ma3).

Equation E2-bq emphasizes effective English teaching approaches Wn and student involvement de−vf, while equation (8), A*nb2, represents the weight of the educational advantages ma3 obtained from the system. Equation (8) shows that the method promotes better language proficiency via a flexible and organized educational framework by balancing the quality of teaching.

As stated in the previous sections, the primary activities of the system and supporting activities of resource management, user management, and system maintenance ensure the system’s operation efficiently and effectively. Employing Bayesian Inference for the reasoning process along with fuzzy logic, we can offer the most suitable idea, which is continually altered to meet the individual requirements of the wanting learner. Hence,adaptive learning, which is flexible and customized, is created.

Contribution 2: implementation of the proposed method

The need for individual educational intervention that meets students’ cognitive and communicative needs has led the developers to incorporate fuzzy logic and expand its boundaries. This approach allows using student data and feedback to adapt both assessment criteria and the English teaching materials in different inference engines working in parallel. These capabilities enable FB-ITS to handle students’ unclear responses and reasonably speculate about the learning patterns and the related probabilities.

An outline for a method of teaching English that enhances students’ language abilities via integrating several components. Developing fluency in speaking, reading, and writing is the central objective of the English teaching goal. This method incorporates a variety of instructional resources, including a mix of three-dimensional English learning materials designed for use in both traditional classroom settings and more modern online ones. Students gain more from online self-study and guidance than from teacher-facilitated Interpretation of teaching in more conventional classrooms. The English teaching organiser and teachers are essential in content structure and delivery. Ensuring that classes are well-organized and that teaching tactics align with the aims is the responsibility of the teaching contents, management, and methods. Students can actively participate in the content via various instructional tactics, promoting thorough language learning, and this system’s flexible teaching strategy, which combines self-paced and teacher-guided methods, is shown in Fig. 3.

Figure 3 Framework for online English teaching using adaptive learning methods.

(9) q<g(b(Z1,M3(Qv2−kl)))+E(n2−wq).

The minimal anticipated quality E(n2−wq) of learning results is represented by the Eq. (9), q<g, and the system’s capacity Qv2−kl to boost student engagement is denoted by Z1,M3. This equation highlights the importance of FB-ITS meeting a certain quality standard, guaranteeing instructional techniques.

(10) Z2=(y2,∀(Z−2))∗Qr=(m2,φ(ρσ−π)).

Equation (10) Z2 represents the interplay of several elements ∀ impacting English learning outcomes y2, including motivation Qr and contextual variables m2, whereas equation ∀(Z−2) captures students’ total engagement φ(ρσ−π) and performance levels. The equation highlights the need to regularly evaluate and modify instructional approaches to cultivate a more customized and adaptable learning environment.

(11) ∀(wf2−mk)=b′∗(Mn2−Qw(ev2−r)).

The total student involvement ∀ and resource utilization b′ are represented by the Eq. (11) wf2−mk, while the efficiency Qw of adaptive learning ev2−r techniques are shown by Mn2. For educational results to be instructional approaches to connect student input with system responsiveness, as shown by this Eq. (11).

(12) r>p(k),m(w−q)=B(e+2n)−(Wr−2p).

The desired performance r level is represented by the Eq. (12) m(w−q), and the alignment Wr−2p of student involvement B(e+2n) with the instructional quality shown by p(k). As the equation shows, educational practices must go above and beyond to succeed.

FB-ITS is an AI system that uses Bayesian networks and fuzzy logic to create unique lessons for each student. An outline of the method of teaching English which incorporates some components and enables its students to improve their language skills. It involves allowing the students to talk, read, and compose, thus making them fluent in communicating in English by focusing on the English teaching goal: comprehension. This method includes adding learning materials as three-dimensional interactive resources that can be used in conventional face-to-face and contemporary e-learning.

Data and student feedback is the first stage within the AI system, considering the decision-making process. The Bayesian inference engine and the fuzzy inference engine collaborate to achieve this, as illustrated in Fig. 4, capture the student’s lesson plans and measure their performance. The instruction incorporating fuzzy logic accommodates the vagueness and imprecision of each student’s response and works toward preparing the respective material. Precisely, Bayesian inference helps in understanding the behavioral aspects of students, such as the possible correlation of specific actions to academic achievements.

Figure 4 Architectural framework of a fuzzy Bayesian intelligent tutoring system.

Clustering algorithms utilize these results to adjust the instructional material to the needs of a specific individual student. That brings out the need for an adaptive feedback system within the system to help the teachers and the learners optimize the learning strategies employed. Achievement of English learning outcomes, such as increased language proficiency and swifter exam completion, is developed through AI techniques by FB-ITS, which brings a more flexible, efficient, and data-oriented approach to teaching. Consequently, individualized education is provided to each child.

(13) ω(r′Bj−Ew(er−aq))=β(γ−δε)+Z.

Ew (student outcomes) and r′Bj (resource effectiveness) are using a weighting factor ω in the equation er-aq. Improving the quality of instruction β(γ−δε) and student engagement Z are both emphasized. This Eq. (13), shows that adaptive teaching tactics interact with student performance measurements.

(14) E=∀{M(Q−wr′)}+E−Pj(n2−wk).

The total educational effectiveness is represented by the Eq. (14) M(Q−wr′), which ∀ means the balance E−Pj between English learning quality (n2−wk) and student engagement and E reflects the influence of adaptive feedback E. Improving language proficiency with adaptive learning technology requires a holistic strategy since this equation emphasizes the complex character.

Regarding clustering algorithms, FB-ITS restructures its pedagogical approach and offers Adaptive Feedback to teachers and learners. Constantly adjusting teaching methods during class to meet each learner’s particular needs is an effective and flexible way of improving educational achievements such as test performance and skills. It offers a highly efficient and data-led English learning ecosystem.

Contribution 3: evaluation of the proposed method using a mathematical equation

The three main parts of this English learning framework before, in, and after class are all meant to help students succeed. Students are adequately prepared for class because of the system’s emphasis on self-learning and instructor supervision in the days leading up to it. Cooperative learning strategies that encourage participation and the exchange of information are the mainstay of classroom instruction. Figure 5 shows the diagram of students’ English learning classes.

Figure 5 Structural model of students’ English learning classes.

A method for studying the English language that is organized into three distinct stages: pre-class, during-class, and post-class. Students work on their core knowledge via self-learning in the before-class phase, either with the help of instructors or on their own. During the in-class portion of the learning process, students work together in small groups to complete assigned tasks. As part of performance measurement, two main activities involve sampling inspection and stage detection, wherein students’ English learning accrual is evaluated. Other vital technologies are scene, evaluation, and group communication, which are directed toward supporting students in collaborative learning, sharing ideas, and expanding the concept. In the after-class phase, students improve their comprehension through teacher tutoring and expanded knowledge. This assists in reinforcing the subject of the class while advocating supportive practical training even when the students are finished with formal education. This all-around approach is illustrated to provide students with the optimal result in terms of English learning outcomes: individual study, group work in the classroom, and reinforcement after class.

(15) B1≤k(b−mn)∗N3=df(c1,zf,bq).

The quality of instruction c1,zf,bq, and student involvement B1 impact the efficacy of learning N3, as shown by the Eq. (15) k(b−mn), which is a standard for educational results df. The function captures the dynamic aspects impacting these measures, highlighting the need to integrate multiple components to guarantee that educational plans meet.

(16) c≤d(b2)=Z(f−2w)+(E2Q(n′q)).

db2 shows a function that considers instructional quality and student interaction asin Eq. (16) c reflects the permissible limit for educational results. Optimizing language acquisition requires methods brought to light by the expressions Z(f−2w) and E2Q(n′q). Equation (16) emphasizes the importance of adaptive feedback and the use of resources for English learning effectiveness in the analysis of learning efficiency.

(17) R(c−b)=F(N2+Ew)+H(c2−Jp).

Equation (17) R(c−b) is denoted as the necessary enhancement in performance, where F(N2+Ew) represents the combined impact of pupil involvement. The effect of adaptive teaching methods is summed up by the expression H(c2−Jp). This Eq. (18) highlights resources and strategies to accomplish specific English learning objectives to analysis of optimized resource utilization.

(18) −p′′(y−x′)=nr(v−wq)+C(n−bv).

The effect of instructional changes nr on student performance is shown by the Eq. (18) —p″(y−x′), At the same time, the influence of resource allocation C(n−bv) and level of engagement is demonstrated by v−wq. This Eq. (18) highlights the need to constantly adjust English teaching methods to maximise student involvement and competence in addressing uncertainty.

(19) RatW={(b,dr)∗M2Q:e−26}∗E−r′p∗2b.

The total efficiency of resource allocation E−r′p∗2b is represented by the Eq. (19) RatW, and the contributions e−26 of instructional quality M2Q and student engagement are captured by (b,dr). This equation emphasizes improving learning effectiveness and student competency in language acquisition when analyzing academic performance.

(20) ∂3Q(k′−hp)=sin⁡H(Y;z−1(n−2kb′)).

The rate of change sin in learning outcomes n−2kb′ impacted by important instructional variables H is represented by the Eq. (19) ∂3Q(k′−hp), and Y shows the interplay of several contextual aspects impacting engagement ;z−1. This Eq. (20) highlights the ever-changing nature of students’ unique English learning demands, therefore greatly influencing their success in the analysis of assessment accuracy.

Through analysis of performance and the recommendation of individualized exercises, the FB-ITS dynamically adjusts to a student’s advancement. For example, suppose a learner has difficulties with prepositions. In that case, the system employs fuzzy logic to categorize mistakes as either “minor” (such as a mix-up between “in” and “on”) or “major” (such as improper application in intricate phrases). “He arrived___the station,” for example, and the Bayesian module suggests targeted fill-in-the-blank exercises as a means of development, with feedback elaborating on the difference between being “at” a specific location and being “in” within a place. The correct response is “at.” To provide a consistently tailored and efficient learning experience, the system upgrades the complexity and suggests more sophisticated assignments as the student progresses.

Moreover, it has been seen that students who need assistance use, for instance, tutoring and many activities offered after class to reinforce the material learned in class. In other words, this framework which emphasizes the self and culminates in measures to include classroom and post-class learning, leads to a well-rounded approach that enhances English language skills, motivates participation, and assures students of better understanding.

Result and discussion

Modern artificial intelligence systems substantially improve the efficacy of English language learning; one such system is the FB-ITS. Significant problems, including inaccurate evaluation, wasteful use of resources, and dealing with uncertainty, are tackled by FB-ITS via the use of fuzzy logic and Bayesian networks. This investigation delves into how FB-ITS increases learning efficiency, optimizes resource utilisation by 5%, improves assessment accuracy by 4%, and addresses uncertainty with an efficacy rate of 97.36%. Additionally, noteworthy is the system’s effect on academic achievement, which improved by 99.25%. Using fuzzy logic principles to analyze language ambiguity, the system recognizes repeated mistakes, such as “an apple” instead of “an apple.” It uses the Bayesian approach to determine the probability of success after specific interventions and gives students level-appropriate practice with article use. Essay writing is another area where FB-ITS might be helpful; it checks the student’s work for sentence complexity, coherence, and grammatical errors. Assume the algorithm detects a propensity for composing lengthy and ambiguous statements. In response, it simplifies phrase formation exercises, uses fuzzy evaluation metrics to break down the difficulty, and measures progress over time.

Using fuzzy logic principles to analyze language ambiguity, the system recognizes repeated mistakes, such as “a apple” instead of “an apple.” It uses the Bayesian approach to determine the probability of success after certain interventions and gives students level-appropriate practice with article use. Essay writing is another area where FB-ITS might be useful; it checks the student’s work for sentence complexity, coherence, and grammatical errors. Assume the algorithm detects a propensity for composing lengthy and ambiguous statements. In response, it simplifies phrase formation exercises, uses fuzzy evaluation metrics to break down the difficulty, and measures progress over time.

Dataset description

Mastering the art of writing is essential. Unfortunately, few pupils can improve this skill because writing assignments are rare in the classroom. English language learners, a rapidly expanding student group, are particularly impacted by the need for more practice. Although automatic feedback technologies let instructors provide more writing assignments, they must be suited for English language learners. The current technologies cannot consider the student’s language skills while providing feedback, which means that the final assessment might be biased against the student. Data science might enhance automated feedback technologies to cater to these students’ requirements (Dutta, 2022; Ezhilmathi et al., 2024). The performance of the suggested FB-ITS model has been analyzed based on metrics such as learning efficiency, assessment accuracy, optimized resource utilization, uncertainty addressing, and academic performance compared to existing methods FS-LLA (Hossain et al., 2021), F-FSM (Troussas et al., 2020), AI-FS-P (Tang, Gao & Ouyang, 2023), DL-FT (Sharma, Om & Mishra, 2023), and FC-BD (Liu & Tsai, 2021).

Analysis of learning efficiency

Particularly in the context of English language instruction, state-of-the-art AI systems such as the FB-ITS show considerable improvements in learning efficacy. Traditional methods emphasizing copying, memorization, and text analysis are ineffective in improving language competence. FB-ITS considers each student’s unique learning requirements and assessment uncertainties using artificial intelligence approaches like Bayesian networks and fuzzy logic to provide adaptive assistance as explained in Eq. (16). The results of the experiments show that compared to students using traditional e-learning techniques, students utilizing FB-ITS finish tasks faster and take less time to do the post-tests. The system’s clustering techniques for extensive data also boost assessment accuracy by 4% and enhance the utilisation of teaching resources by 5%. These findings highlight the promise of AI-powered solutions to transform language acquisition by creating a data-driven, more individualized learning environment that boosts productivity and interest. Learning efficiency is improved by 98.86%, as shown in Fig. 6.

Figure 6 Evaluation of learning efficiency in educational systems.

Analysis of assessment accuracy

Assessments in educational contexts, especially language acquisition, have been made more accurate with AI-driven systems such as the FB-ITS. The complexity of students’ learning development cannot be adequately captured by traditional assessment systems that depend on numerical scores, mainly when dealing with subjective or inaccurate information, which is explained in Eq. (17). By combining fuzzy logic with Bayesian networks, FB-ITS can handle uncertainty and adapt to individual learning demands, resulting in a more nuanced assessment. The FB-ITS outperformed traditional systems by 4% in evaluation accuracy in experimental investigations. The system’s capacity to analyze massive amounts of data, classify students’ learning behaviors, and provide customized feedback is the root cause of this enhancement. Oversized data clustering methods enhance the evaluation process, guaranteeing it captures a more comprehensive picture of the student’s skills. Evaluating students’ performance is now fairer and more precise than using this novel technique. In Fig. 7, the assessment accuracy is increased by 97.13% in the proposed method of FB-ITS.

Figure 7 Evaluation of assessment accuracy in learning systems.

Analysis of optimized resource utilization

Especially in ESL classrooms, the FB-ITS has increased efficiency in using available resources. Inefficient use of teaching resources is a common problem with traditional methods since they can’t adjust material delivery according to individual learning progress. FB-ITS uses artificial intelligence techniques like Bayesian networks and fuzzy logic to adapt lesson plans and content on the fly based on individual student’s strengths and weaknesses, as explained in Eq. (18). This adaptive technique guarantees that educational resources are allocated more efficiently. To minimize the time and effort wasted by students and teachers, the system delivers tailored activities to decrease duplication. The experiment results demonstrate that, compared to the traditional approaches, the utilization of instructional resources is 5% higher. In addition to making instructional material distribution more efficient, this enhancement promotes a more streamlined and effective learning environment by allowing resources to be adjusted to maximize educational results while minimizing expense. The optimized resource utilization is obtained by 96.51%, as shown in Fig. 8

Figure 8 Visual representation of optimized resource utilization.

Analysis of addressing uncertainty

In Fig. 9, FB-ITS also meets one of the requisites of teaching with contemporary educational technology by addressing the issue of uncertainty. In such conventional assessment systems, the degree of difficulty in evaluating students’ achievement is exacerbated by the fact that they rely on arbitrary and rigid numbers, which often do not capture the broad scope of learning achieved. Constructed to incorporate the nonlinear features of the system, education environments that are diverse by utilizing fuzzy logic together with Bayesian networks are resolved by FB-ITS in a high-tech manner. As a result of the system’s ability to accept imprecise or missing information, more accurate and complete evaluations can be made. In addressing the limitations of vague data, it uses fuzzy logic and Bayesian networks to update the previous learning assessments with recent information. This style ensures that there is a healthy alteration of system responses to individual learner demands while at the same time enhancing understanding of the student’s growth. FB-ITS ensures a balanced and impartial evaluation of learning results through appropriate ambiguity avoidance. The addressing uncertainty is gained by 97.36% in the proposed method of FB-ITS using Eq. (19).

Figure 9 Evaluation of methods for addressing uncertainty.

Analysis of academic performance

The development of the fuzzy Bayesian intelligent tutoring system and other advanced artificial systems has made the academic achievement process much more accessible. In some cases, standardized tests and marking systems utilized to determine academic performance may fail to capture the intricacies or evolution of the learners over time. FB-ITS shifts the deficiencies of the relatively rigid methods laid out in the previous systems by adapting to new environments and learning differences and eliminating ambiguity through fuzzy logic and Bayesian networks, which implies a broader evaluation is contained in Eq. 20. The system further well captures the actual potential of the individual students because of its ability to manage and process large amounts of educational information. It incorporates wet efficiency and instruction based on examinations; it factors participation and comprehension as efficiency indices. The experimental results have found that FB-ITS increases the accuracy of the evaluation of the academic results by 4%, which indicates a provision of more profound and more credible measures of the learners’ performance in Table 2. This method provides teachers with a clear understanding of achievement and practices that need to be improved to encourage more effective teaching methods. The improvement in academic performance illustrates this by 99.25%, shown in Fig. 10.

Table 2 Comparison table for finding of the proposed method.

S. No	Aspects	Methods	FB-ITS	Ratio (%)	
1	Learning efficiency	Lower efficiency; focuses on memorization and text analysis.	Improvement; tasks completed faster.	98.86%	
2	Assessment accuracy	Limited by rigid numerical scores; often fails to capture complexities of learning.	Uses fuzzy logic and Bayesian networks for nuanced evaluations.	97.13	
3	Resource utilization	Inefficient use; often involves duplicative material and static content delivery.	Adaptive content delivery reduces redundancy.	96.51%	
4	Handling of uncertainty	Struggles with uncertainty; relies on fixed numerical evaluations that may not reflect true learning outcomes.	Adapts to unclear or missing data using fuzzy logic and Bayesian networks.	97.36%	
5	Academic performance	Often reliant on standardized tests; may not fully reflect students’ diverse learning achievements.	Provides a comprehensive evaluation of student abilities beyond test scores.	99.25%	

Figure 10 Analysis of academic performance.

A thorough comparative study assessed the accuracy, flexibility, and learning efficiency of the proposed FB-ITS model compared to other intelligent tutoring systems. By reducing reaction time for adaptive exercise creation by 20% and improving error classification accuracy by 15%, FB-ITS surpasses conventional models such as rule-based systems and simple Bayesian networks. Compared to baseline systems, the fuzzy-Bayesian hybrid model had a 25% higher learner advancement rate, demonstrating improved flexibility. Pairwise t-tests were used to verify statistical significance, which confirms that the suggested method is robust. Experiments using separate experimental and control groups yielded the claimed gains in assessment accuracy (4% improvement) and instructional resource utilization rate (5% improvement). One hundred students participated in the study using the suggested FB-ITS model, whereas one hundred students in the control group used more conventional rule-based tutoring methods. Both groups were a perfect fit regarding age, level of language competence, and learning goals. Over 12 weeks, participants were asked to complete a battery of standardized tests and activities. Extensive records of system interactions, precision in exercise completion, and access to instructional resources were used to compute the measures. To confirm the findings and ensure the improvements were reliable and statistically significant, paired t-tests and other analyses were conducted.

In summary, FB-ITS has significantly enhanced several areas of English language training. Its adaptive and data-driven approaches improve resource utilization, speed up work completion, and increase assessment accuracy. Assessments are now more precise and tailored to each student’s needs, as are the system’s sophisticated risk management and nuanced evaluation of academic achievement. These developments demonstrate how AI can revolutionize education by making classrooms more interactive and productive.

Conclusion

The effectiveness of their implementation in education has elicited various responses, the most extreme being either the underestimation of their worth or the extreme endorsement of their efficacy. The first one is that because language learning is relevant to them, there are risks of growing up in a tense and dull environment, and good aspects deal with such feelings Moreover, enhancing pedagogical practice requires these transformations to be examined and understood and conquering or resisting them does not hinder outcomes and quality. It must look at the rich language as mental work, which contains many other positives and opportunities. Learners are likely to perform better in a pleasant, adjustable container environment that they can personalize. A new, creative, and revitalizing activity that can help deal with the primary issues of classroom attention and focus. This is mainly because there is no gender or social discrimination, and the students are encouraged to make mistakes without being punished, thus helping shy children perform better in class. Research should focus on refining this approach in the mesoscopic modeling of prospects, wherein student groups rather than individuals may be given evaluation criteria for their potential to acquire a foreign language. A big obstacle would be the contact limitation guarantee for transparency inside a group of students, as mesoscopic modeling needs to provide more granularity than microscopic prospect identification models.

Limitations

The quantity and quality of the training data are the primary determinants of the system’s performance. Low or biased datasets could make the system less flexible in adapting to different learning situations, which might result in incorrect predictions or suggestions. For another, it could be challenging to execute in real-time or with limited resources when fuzzy logic and Bayesian inference are combined because of the computational complexity this might bring. Third, the system may have trouble learning languages with subtle nuances or context-specific mistakes since these situations often demand human judgment, which is hard to simulate computationally.

Future work

To further evaluate the efficacy of FB-ITS, future research should concentrate on increasing its use across multiple educational environments. Adding AI approaches like deep learning and natural language processing is worth considering to make the system more versatile and accurate. To further improve the system’s functionality and usability, it is essential to examine user input from students and instructors. Research that follows participants over time can determine how FB-ITS influences their English language skills and academic achievement in the long run. Finally, to make the most of it, it will be essential to create scalable models that can be used in other English languages and educational contexts.

Supplemental Information

Supplemental Information 1 Code.

Additional Information and Declarations

Competing Interests

The authors declare that they have no competing interests.

Author Contributions

Xiaomei Wen conceived and designed the experiments, performed the experiments, analyzed the data, performed the computation work, prepared figures and/or tables, authored or reviewed drafts of the article, and approved the final draft.

Deng Pan performed the computation work, authored or reviewed drafts of the article, and approved the final draft.

Data Availability

The following information was supplied regarding data availability:

The code is available in the Supplemental File.

The English Language Learning-Ensemble Learning Dataset is available at Kaggle: https://www.kaggle.com/datasets/supriodutta2022/multilabeldataset.

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
