# Peer review of "Charting new territories: fuzzy systems in English language teaching and learning"

_PeerJ Computer Science, doi:10.7717/peerj-cs.2887_

## Round 0.1 · original submission · Major Revisions

According to the three reviewers' comments and suggestions, this paper needs major revisions.

Reviewer 1 ·

Basic reporting

English can be improved

References are sufficient

Figure quality can be improved, raw data is not shared

Experimental design

For me the paper can be improved with some examples

Validity of the findings

no comment

Additional comments

This paper investigates the potential of the Fuzzy Bayesian Intelligent Tutoring System (FB-ITS), an AI-based system utilizing fuzzy logic and Bayesian networks, to enhance English language teaching and learning. The authors identify challenges in current pedagogical approaches, including over-reliance on traditional methods that do not prioritize listening and speaking skills. The study highlights the limitations of existing evaluation methods and proposes an innovative, adaptive tutoring system to address these issues. Experimental findings demonstrate the system's efficiency and effectiveness, with notable improvements in assessment accuracy and teaching resource utilization.

The topic is compelling, and the paper provides a robust literature review. However, there are areas that need refinement, including formatting issues, clarity in presenting results, and the choice of supplementary materials, which could improve the manuscript's professional and academic quality.

List of detailed comments as follows:
- Page 7, line 82: The word "i" should be moved to the next line with "improve" in the first point.
- Why are the figure captions written in all capital letters?
- Equation (5) is not centered and should be adjusted for proper alignment within the manuscript.
- References to equations in the text are improperly formatted. For instance, on pages 14 and 15, terms like Rc-b and db2 are used instead of the consistent format "(16)" and "(17)." Similar issues are on page 11, such as the description of "(8)."
- There are no figures labeled 6a and 6b.
- Including screenshots of Python code seems unprofessional and unrelated to the study's focus. It is recommended to remove these from the paper.
- To enhance the paper's utility and relevance, add screenshots of the implemented interface system, examples of exercises, or details of how the recommendation system works, supplemented with practical examples.

Cite this review as

Reviewer 2 ·

Basic reporting

This paper studies Charting New Territories: Fuzzy Systems In English Language Teaching And Learning. It is interesting. Some comments are provided as follows.

In section 3, some symbols lack explanation. Any symbol need to be clarified when it appears for the first time.
All equations in this paper should be rewritten strictly according to the mathematical form.

There is no any citation in section 3. It seems that this paper just uses existing FB-ITS system to do some tests. It is needed to clarify the novelty and scientific contribution.

Please add some solid comparative analyses in section 4 to strengthen the convincingness and credibility of this paper.

In sum, I recommend major revision.

Experimental design

Please add some solid comparative analyses in section 4 to strengthen the convincingness and credibility of this paper.

Validity of the findings

In section 3, some symbols lack explanation. Any symbol need to be clarified when it appears for the first time.
All equations in this paper should be rewritten strictly according to the mathematical form.

There is no any citation in section 3. It seems that this paper just uses existing FB-ITS system to do some tests. It is needed to clarify the novelty and scientific contribution.

Please add some solid comparative analyses in section 4 to strengthen the convincingness and credibility of this paper.

Additional comments

This paper studies Charting New Territories: Fuzzy Systems In English Language Teaching And Learning. It is interesting. Some comments are provided as follows.

In section 3, some symbols lack explanation. Any symbol need to be clarified when it appears for the first time.
All equations in this paper should be rewritten strictly according to the mathematical form.

There is no any citation in section 3. It seems that this paper just uses existing FB-ITS system to do some tests. It is needed to clarify the novelty and scientific contribution.

Please add some solid comparative analyses in section 4 to strengthen the convincingness and credibility of this paper.

In sum, I recommend major revision.

Cite this review as

Reviewer 3 ·

Basic reporting

In this paper the author developed The Fuzzy Bayesian Intelligent Tutoring System (FB-ITS), based on fuzzy logic and the Bayesian network methodologyis. This AI system supports students in English Teaching and learning settings.
The topic is interesting, however, I think that certain missing segments need to be significantly improved.

Experimental design

My comments are as follows:
1. Explain in detail why you chose The Fuzzy Bayesian Intelligent Tutoring System, why you did not choose some other models, such as only FIS, ANFIS, or neural network, etc. Elaborate in detail what was decisive for the authors to create such a model.
2. The presentation of results and discussion should be more detailed.
3. Form a separate section to deal with model limitations.

Validity of the findings

My comments are as follows:
1. How confident are we in the results of the method? What about a detailed comparative analysis with previous models? Create a separate section related to comparative analysis and show that your model is better than others.
2. The authors state: "The assessment accuracy has risen by 4%, and the teaching resource utilization rate has risen by 5%." However, it should be explained in detail how all the data was obtained. The compositions of experimental groups and control groups should be described in detail when testing the model.

Additional comments

My comments are as follows:
1. In table 1 should list the sources for each model.
2. The literature analysis is based on older researches. You should extend the literature review. Analyze new manuscripts, period 2023-2025. years
3. The conclusion section seems to rush to the end. The authors will have to demonstrate the impact and insights of the research. The authors need to rewrite the entire conclusion section with focus on both impact and insights of the manuscript. Clearly state your unique research contributions in the conclusion section.

Cite this review as

---

## Round 0.2 · Major Revisions

Based on the reviewers' comments and suggestions, this article still needs major revisions.

Reviewer 1 ·

Basic reporting

I feel that paper was not improved at all, despite rebuttal letter claiming that it was. Some of the suggested/requested suggested were simply ignored, but authors claim that they made this. The paper is in wrong version or the authors didn't really improve much.

Rev1 - Answer 3 - "Figure quality improved" - I didn't notice any significant difference between old and new version of the figures.

"All figure captions have been modified" - Nothing changed

"Equation (5) has been centered" - Nothing changed

"Python code has been removed" - Nothing changed

Requested examples was provided in the text of the rebuttal, but not in the paper.

"Table 1 soruce has been provided" - No? Or it is unclear where the authors put it.

Experimental design

No comment

Validity of the findings

No comment

Additional comments

No comment

Cite this review as

Reviewer 2 ·

Basic reporting

This paper still needs some minor revisions.
In section 1, the second contribution is not complete.
In section 4, please divide it into several subsections and clarify comparisons with existing methods in each subsection.

Experimental design

This paper still needs some minor revisions.
In section 1, the second contribution is not complete.
In section 4, please divide it into several subsections and clarify comparisons with existing methods in each subsection.

Validity of the findings

This paper still needs some minor revisions.
In section 1, the second contribution is not complete.
In section 4, please divide it into several subsections and clarify comparisons with existing methods in each subsection.

Additional comments

This paper still needs some minor revisions.
In section 1, the second contribution is not complete.
In section 4, please divide it into several subsections and clarify comparisons with existing methods in each subsection.

Cite this review as

---

## Round 0.3 · accepted · Accept

Based on the reviewers' suggestion, the revised version can be accepted for publication.

Reviewer 1 ·

Basic reporting

Ok

Experimental design

Ok

Validity of the findings

Ok

Additional comments

The authors have done required improvements therefore I think this paper can be accepted.

Cite this review as

Reviewer 2 ·

Basic reporting

It can be accepted now.

Experimental design

It can be accepted now.

Validity of the findings

It can be accepted now.

Additional comments

It can be accepted now.

Cite this review as

Reviewer 3 ·

Basic reporting

The authors did not remove the objections from the previous round of review.

Experimental design

The authors did not remove the objections from the previous round of review.

Validity of the findings

The authors did not remove the objections from the previous round of review.

Additional comments

The authors did not remove the objections from the previous round of review.

Cite this review as